# On the Distribution of Magnetic Moments in a System of Magnetic Nanoparticles

**Max Javier Jáuregui Rodríguez [1], Denner Serafim Vieira [1], Renato Cardoso Nery [1], Gustavo Sanguino Dias [1]**, **Ivair Aparecido dos Santos [1]**, **Renio dos Santos Mendes [1,2] and Luiz Fernando Cotica [1,*]**

[1] Department of Physics, State University of Maringá, Avenida Colombo, 5790, Maringá 87020-900, Paraná, Brazil
[2] National Institute of Science and Technology for Complex Systems, Rua Xavier Sigaud 150, Rio de Janeiro 22290-180, Rio de Janeiro, Brazil
[*] Correspondence: lfcotica@dfi.uem.br

**Abstract:** Particle size distribution carries out a substantial role in the magnetic behavior of nanostructured magnetic systems. In fact, a vast literature on superparamagnetism has been reported, suggesting that the particle size distribution in a system of magnetic nanoparticles (MNPs) corresponds to a lognormal probability density function, and several works have properly considered their magnetic moments following a similar distribution, as a universal rule. In this manuscript, it is demonstrated that alternative probability distribution functions, such as the gamma and Weibull ones, can be used to obtain useful parameters from the analysis of the magnetization curves, indicating there is no universal model to represent the actual magnetic moment distribution in a system of magnetic nanoparticles. Inspired by this observation, a reliable method to properly identify the actual magnetic moment distribution in a given nanostructured magnetic system is proposed and discussed.

**Keywords:** magnetic nanoparticles; magnetic moment distributions; distribution models

## 1. Introduction

Magnetic nanoparticles (MNPs) have been used in numerous technological applications. A variety of these particles, composed of distinct atoms or ions with different magnetic moments, have been synthesized to be applied in many research fields. For instance, inorganic MNPs such as $Fe_3O_4$ (magnetite) have been widely employed in biophysical applications [1,2]. In biomedicine, water or alcohol dispersible MNPs have created new opportunities for diagnostic imaging [3–8], tumor treatment via hyperthermia [9,10], biomolecular separation and magnetic field-controlled drug delivery [11–13]. In the latter, drug transport through MNPs has been extensively studied in an attempt to obtain particles with high drug carrier capacity, good biocompatibility with cells and tissues, and good stability in aqueous solutions [14].

Most of the applications of MNPs rely on the fact that they show a superparamagnetic behavior, having the main advantage that they do not easily agglomerate, in contrast to ferromagnetic particles. That particular behavior can improve the performance and stability of the fluid used in the treatments previously mentioned. Although many studies focus on practical applications of MNPs, one of the main problems is to understand their magnetic behavior at nanometric scales.

Usually, the magnetic moments and volumes in an assembly of MNPs are distributed due to, for example, the synthesis procedure. Accurate knowledge of the particle size and magnetic moment distributions as well as the mean magnetic moment in an assembly of MNPs can be crucially important for the proper working of a particular application. As superparamagnetic nanoparticles at high temperatures become single-domain nanoparticles [15], usually, the magnetic moment and the size of each particle are correlated. In this

direction, the magnetic moment distribution has been associated with the particle size distribution in systems of MNPs.

Since the early studies, the search for the particle size distribution based on magnetization measurements was mainly conducted in ferrofluids [16–19]. The result of these measurements combined with the particle size distribution obtained by electron microscopy led researchers to use the lognormal probability density function (lognormal PDF). This PDF can be typical for particles formed by a grinding process, a well-known top-down process that tends to produce particles with a broad distribution of sizes [20].

In addition to the grinding processes, bottom-up methods have been proposed to synthesize magnetite nanoparticles [21–28]. In general, these procedures are wet chemical routes and the most used of them is based on the co-precipitation of $Fe^{2+}$ and $Fe^{3+}$ aqueous salt solutions by addition of a base [21]. The size, shape, and composition of synthesized nanoparticles depend on the type of used salts (chlorides, sulphates, nitrates, etc.), $Fe^{2+}/Fe^{3+}$ ratio, pH and ionic strength of the media [22,23].

Despite the majority of synthesis routes being bottom-up, many researches are still using a lognormal PDF to describe the particle size and magnetic moment distributions in MNPs assemblies as magnetite [16,17,29–33] and maghemite [18,34–36] . However, it can be found in the literature works using mean value calculations [31,37–40] or bimodal [41,42], Gaussian [17,29] and gamma [29] PDFs to model the particle size distribution, among other models [43,44].

In this paper, we discuss the limitations of the description of a magnetic moment distribution in a system of MNPs using a model based on the lognormal PDF. In addition, we propose other approaches to the study of this problem; for instance, the use of models based on mean value calculations or gamma and Weibull PDFs instead of the lognormal one. Furthermore, we investigate a method for identifying the actual magnetic moment distribution in a system of MNPs. The organization of the article is as follows: Section 2 shows the fittings of the magnetization curves of some samples using models that involve the lognormal, gamma and Weibull PDFs; Section 3 describes a possible method for obtaining the actual magnetic moment distribution of an ideal system of MNPs; finally, we conclude in Section 4.

## 2. Fittings to the Magnetization Curves

A common practice in the study of the magnetic moment distribution in a system of superparamagnetic MNPs is to perform a fitting process, where the magnetization curve (curve *M* vs. *H*) is fitted using a model. Consequently, in order to verify the behavior of the models that we will propose in this section, we will fit the magnetization curves of some real samples.

### 2.1. Samples

The magnetite ($Fe_3O_4$) samples used in this work were synthesized via thermal decomposition of a ferric nitrate/ethylene glycol solution. As explained in Ref. [45], first, a mixture of adequate amounts of ferric nitrate and ethylene glycol was prepared. The solution was homogenized at room temperature followed by heating at 90 °C. Then, in order to obtain nanoparticles with different sizes and distributions, the resulting material was heated in a tube furnace at temperatures between 300 °C and 600 °C, under inert atmosphere (argon). The particle diameter sizes are between 15 and 20 nm (similar sizes were calculated via X-ray diffraction measurements). We will use the notation MagT-nh to refer to the sample that was heated at a temperature of *T* °C for *n* hours.

### 2.2. Models

Inspired by a set of noninteracting magnetic dipoles (see Appendix A), we will use the following models in the investigation of the magnetic moment distribution in a system of MNPs:

(M) *Modified Langevin function* [46–51]:

$$M = \chi H + M_S L\left(\frac{\langle\mu\rangle H}{k_B T}\right)$$

(1)

were $\chi$ is a mass magnetic susceptibility, $M_S$ is the saturation magnetization, $L(x) = cothx - 1/x$ is the Langevin function, $\langle\mu\rangle$ is a mean magnetic moment and $k_B$ is the Boltzmann constant.

(L) *Lognormal model* [32,52–54]:

$$M = \chi H + c \int_0^\infty \mu L\left(\frac{\mu H}{k_B T}\right) l_{\lambda,\beta}(\mu) d\mu ,$$

(2)

where $c$ is a constant and

$$l_{\lambda,\beta}(\mu) = \frac{1}{\sqrt{2\pi}\beta\mu} exp\left\{-\frac{[\ln(\mu/\lambda)]^2}{2\beta^2}\right\} ,$$

(3)

is the lognormal PDF with parameters $\lambda > 0$ and $\beta > 0$. Where $\lambda$ is the median core size and $\beta$ gives the width of the distribution.

(G) *Gamma model*:

$$M = \chi H + c \int_0^\infty \mu L\left(\frac{\mu H}{k_B T}\right) g_{\lambda,\beta}(\mu) d\mu ,$$

(4)

where

$$g_{\lambda,\beta}(\mu) = \frac{\lambda^{-\beta}\mu^{\beta-1}e^{-\mu/\lambda}}{\Gamma(\beta)} ,$$

(5)

is the gamma PDF with parameters $\lambda > 0$ and $\beta > 0$. Where $\lambda$ is the median core size and $\beta$ gives the width of the distribution.

(W) *Weibull model*:

$$M = \chi H + c \int_0^\infty \mu L\left(\frac{\mu H}{k_B T}\right) \omega_{\lambda,\beta}(\mu) d\mu ,$$

(6)

where

$$\omega_{\lambda,\beta}(\mu) = \frac{\beta}{\lambda}\left(\frac{\mu}{\lambda}\right)^{\beta-1} exp\left[-\left(\frac{\mu}{\lambda}\right)^{\beta}\right] ,$$

(7)

is the Weibull PDF with parameters $\lambda > 0$ and $\beta > 0$. Where $\lambda$ is the median core size and $\beta$ gives the width of the distribution.

The mass magnetic susceptibility $\chi$ that appears in all models can be associated with the randomly oriented ferrimagnetic particle cores [54]. However, different magnetic orders as ferromagnetic or antiferromagnetic ones can be considered. In fact, the effect can also come from some spin canting effect or paramagnetic contribution to $M$, i.e., the mass magnetic susceptibility $\chi$ can take on the role of different magnetic configurations/states, including some exotic states [55]. For example, it looks like the $M/H$ curve for the MAG300-1h sample is typical for magnetic nanoparticles in a blocked regime. However, we do not have enough elements to conclude that. Thermally blocked magnetic nanoparticles should show hysteresis (remanence and coercivity) but this cannot be seen in the $M/H$-loop for MAG300-1h. The high field susceptibility is not due to thermal blocked properties. Instead, as a hypothesis, the high field susceptibility can be due to, for instance, a large spin canting effect (that may be due to the fact that the particles have a small core size).

The models (M) and (L) have been used in many investigations about systems of MNPs. The wide use of the model (L) is mainly motivated by the fact that, in the literature, the particle size distribution, obtained by microscopy, is usually fitted with a lognormal PDF [16–18,29–32,34–36,56] . However, the particle size distribution could also be fitted

with gamma or Weibull PDFs since the lognormal, gamma and Weibull PDFs can be made similar by conveniently choosing the values of their parameters [57,58]. For this reason, we have introduced the models (G) and (W) which, to the best of our knowledge, were not used in the literature on the study of the magnetic moment distribution in systems of MNPs. Moreover, the connection between particle size and magnetic moment distributions depends on the particles in the assembly being all superparamagnetic, which is not always true. In this work, we will only study the magnetic moment distribution, without discussing its connection with the particle size distribution.

In contrast to the model (M), the models (L), (G) and (W) use a continuous probability distribution for the magnetic moments. Hence, in these cases, the mean magnetic moment will be given by $\langle \mu \rangle = \int_0^\infty \mu f(\mu)\,d\mu$, where $f(\mu)$ stands for the lognormal, gamma or Weibull PDFs, respectively. Moreover, in these cases, it is also convenient to define the saturation magnetization by $M_s = c\langle \mu \rangle$.

*2.3. Fittings*

Fittings to the magnetization curves of the Mag300-1h, Mag300-2h, Mag500-2h and Mag600-2h samples were performed using the models (M), (L), (G) and (W). For each sample, we obtained four very good fits (see Figure 1). The fitted curves are so similar that it is very difficult to visually determine the best curve, and, consequently, to choose the best model. This fact indicates that the models (M) and (L) have nothing special, compared to the models (G) and (W).

Associated with each fitted curve, there are coefficients of determination $R^2$ and an Akaike information criterion (AIC) value which can help us to decide which curve is the best for each sample. If we consider models with the same number of parameters [e.g., (L), (G) and (W)], we just have to evaluate the quality of the models in order to choose the best one. Consequently, the model with the value of $R^2$ closest to one will be the best. However, if we have models with different numbers of parameters [e.g., (M) and (L)], we must take into account the complexity (number of parameters) and the quality of the model. In this case, the AIC plays an important role, since it evaluates the quality of the model and also penalizes its complexity. According to Ref. [59], the best model in that situation will be the one with the lowest AIC value.

Table 1 shows the values of the fitting parameters and also the mean magnetic moment, the coefficient of determination $R^2$ and the AIC value. We notice that, based on the lowest AIC value, the best models for the Mag300- 1h, Mag300-2h, Mag500-2h and Mag600h samples are, respectively, (G), (L), (M) and (W). We can also notice that, for all samples, we obtain better fits ($R^2$ closer to 1) with the models (L), (G) and (W), which use a continuous probability distribution, than with the model (M). This makes the idea of a continuous magnetic moment distribution more appealing in systems of MNPs.

Another interesting fact shown in Table 1 is that the mean magnetic moment may vary greatly depending on the model that is used. This can also be seen from the shape of the PDFs (see Figure 2). For example, for the Mag300-1h sample, there is a large discrepancy between the mean magnetic moment given by the model (G) and the ones given by the other models (there is also a discrepancy in the values of the parameter c). In this situation, we could think that the model (G) is not physically consistent despite it being the best model for the Mag300-1h sample, according to the lowest AIC value. However, this affirmation can be misleading. If we restrict our attention to the facts, the only assertion that we can make is that both the magnetic moment distribution and the mean magnetic moment are not free from ambiguities. This suggests that we should not impose the form of the magnetic moment distribution but find it, at least approximately, directly from the magnetization data. The next section shows an attempt to perform this task.

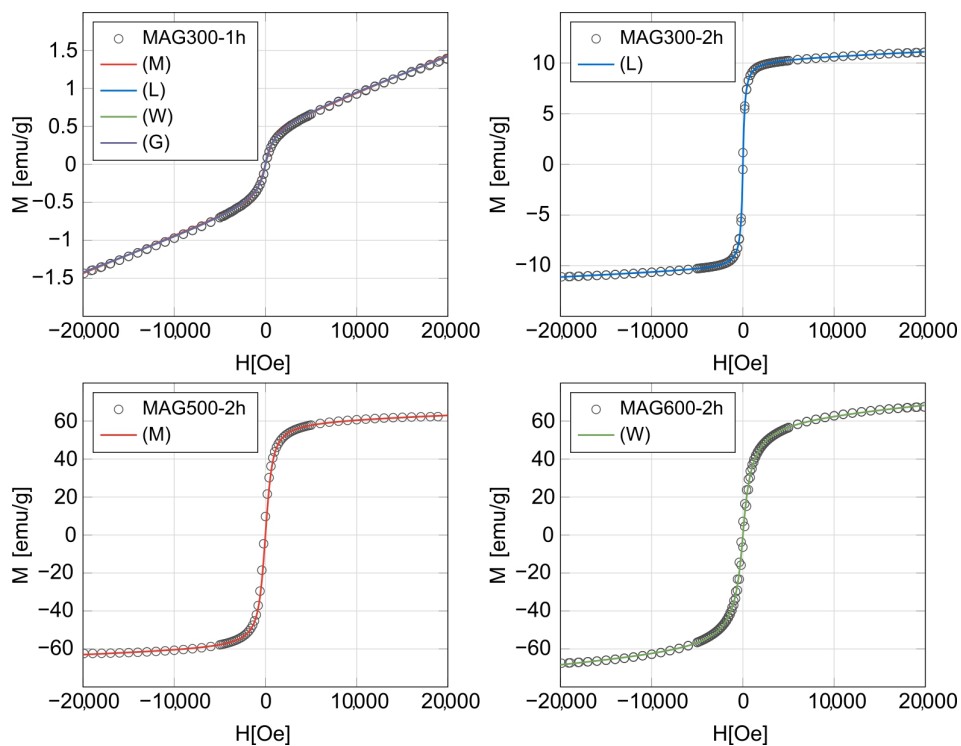

**Figure 1.** Fittings to the magnetization curves of the Mag300-1h, Mag300-2h, Mag500-2h and Mag600-2h samples using the models (M) Modified Langevin function, (L) lognormal, (G) gamma and (W) Weibull. For each sample, the fitted curves are very similar. For this reason we only show the best-fitted curve for the last three samples, i.e., the model which has the lowest Akaike information criterion (AIC) value. Experimental measurements were performed at room temperature (300 K).

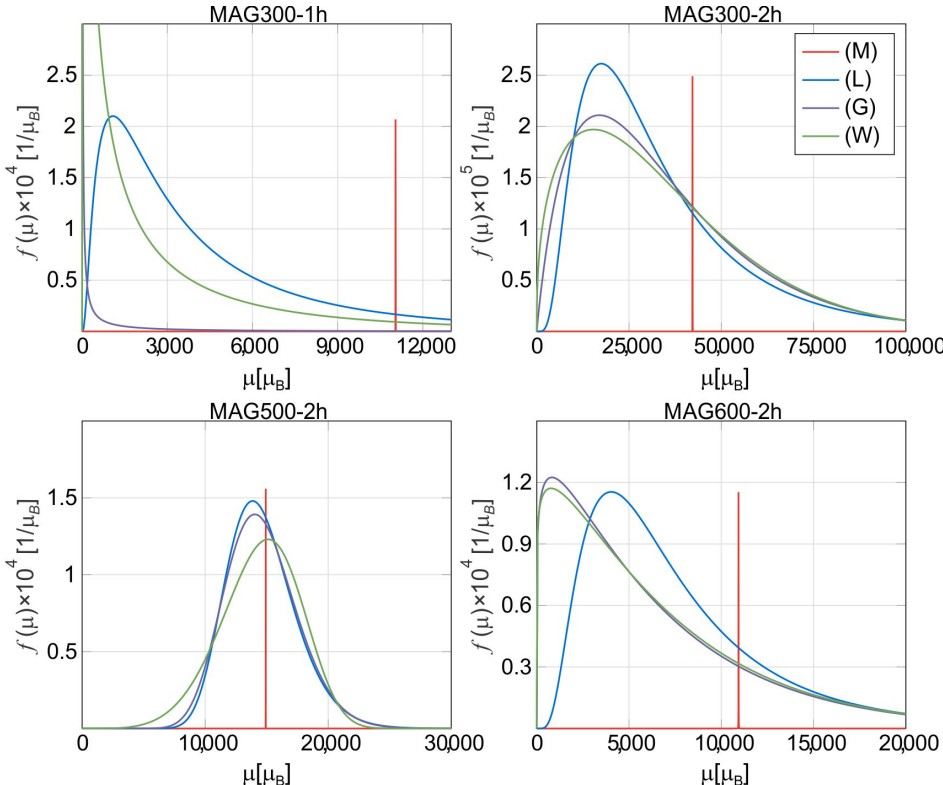

**Figure 2.** Representation of the probability density functions used to fit the magnetization curves of the Mag300-1h, Mag300-2h, Mag500-2h and Mag600-2h samples.

**Table 1.** Parameters of the models (M) Modified Langevin function, (L) Lognormal, (G) Gamma and (W) Weibull obtained by fitting the magnetization curve of the Mag300-1h, Mag300-2h, Mag500-2h and Mag600-2h samples. The mean magnetic moment (in Bohr magnetons, $\mu$B), the coefficient of determination R and the Akaike information criterion (AIC) value are also shown. Although models (L), (G) and (W) do not have an $M_S$ parameter, it is convenient to define $M_S = c\langle\mu\rangle$ in these cases.

| Model | $\chi$ ($\times 10^{-5}$ cm$^3$/g) | $M_S$ (emu/g) | $\lambda$ ($\mu_B$) | $\beta$ | $c$ ($\times 10^{16}$ g$^{-1}$) | $\langle\mu\rangle$ ($\mu_B$) | $R^2$ | AIC |
|---|---|---|---|---|---|---|---|---|
| | | | Mag300-1h | | | | | |
| (M) | 4.9 | 0.47 | | | | 11,042.3 | 0.999133 | $-750.955$ |
| (L) | 4.5 | 0.54 | 3133.78 | 1.03 | 1.09 | 5327.46 | 0.999261 | $-774.827$ |
| (G) | 4.5 | 0.24 | 12,531.1 | 0.02 | 24.0 | 106.96 | 0.999264 | $-775.482$ |
| (W) | 4.5 | 0.56 | 1912.33 | 0.57 | 1.96 | 3062.9 | 0.999264 | $-775.337$ |
| | | | Mag300-2h | | | | | |
| (M) | 5.0 | 10.2 | | | | 42,216.1 | 0.999860 | $-222.430$ |
| (L) | 4.3 | 10.3 | 28,108.2 | 0.69 | 3.12 | 35,634.2 | 0.999898 | $-271.347$ |
| (G) | 4.2 | 10.3 | 17,798.2 | 1.95 | 3.20 | 34,762.0 | 0.999896 | $-269.014$ |
| (W) | 4.2 | 10.3 | 37,355.8 | 1.40 | 3.28 | 34,033.7 | 0.999896 | $-268.319$ |
| | | | Mag500-2h | | | | | |
| (M) | 16.7 | 60.5 | | | | 14,924.9 | 0.998594 | 348.987 |
| (L) | 15.7 | 60.7 | 14,376.2 | 0.19 | 44.7 | 14,640.8 | 0.998595 | 350.913 |
| (G) | 15.6 | 60.7 | 580.66 | 25.2 | 44.8 | 14,614.0 | 0.998595 | 350.908 |
| (W) | 15.2 | 60.8 | 15,757.3 | 5.16 | 45.2 | 14,494.8 | 0.998596 | 350.880 |
| | | | Mag600-2h | | | | | |
| (M) | 63.2 | 57.7 | | | | 10,937.0 | 0.999038 | 628.309 |
| (L) | 45.1 | 61.1 | 6405.28 | 0.68 | 81.6 | 8071.44 | 0.999162 | 608.134 |
| (G) | 39.1 | 62.6 | 5723.98 | 1.14 | 103 | 6545.34 | 0.999176 | 605.332 |
| (W) | 40.1 | 62.3 | 6898.42 | 1.10 | 101 | 6657.45 | 0.999177 | 605.160 |

## 3. Approximation to the Actual Magnetic Moment Distribution of an Ideal Sample

Assuming that there is a continuous particle magnetic moment distribution, which is reasonable, there exists the possibility that this distribution does not correspond to lognormal, gamma or Weibull PDFs. In this case, we could use the following formula for magnetization:

$$M = \chi H + c \int_0^\infty \mu L\left(\frac{\mu H}{k_B T}\right) g(\mu)\, d\mu \, , \tag{8}$$

where $g(\mu)$ is an unknown PDF associated with magnetic moment distribution.

Although in some cases the mass magnetic susceptibility $\chi$ may not follow a $T^{-1}$ law (e.g., it could follow a $T^{-1/2}$ law [52,60]), if this happens, Equation (8) can be obtained as a particular case of the following equation

$$M = \tilde{c} \int_0^\infty \mu L\left(\frac{\mu H}{k_B T}\right) f(\mu)\, d\mu \, . \tag{9}$$

Indeed, if we consider the PDF $f(\mu) = (1-p)\delta(\mu - \mu_1) + pg(\mu)$, where $0 \leq p \leq 1$, $\delta(\mu)$ is the Dirac delta distribution and $g(\mu)$ is a PDF, we obtain

$$\tilde{c} \int_0^\infty \mu L\left(\frac{\mu H}{k_B T}\right) f(\mu)\, d\mu = (1-p)\tilde{c}\mu_1 L\left(\frac{\mu_1 H}{k_B T}\right) + p\tilde{c} \int_0^\infty \mu L\left(\frac{\mu H}{k_B T}\right) g(\mu)\, d\mu \, . \tag{10}$$

Assuming also that $\mu_1 H \ll k_B T$, the first term on the right-hand side of Equation (10) is approximately given by $\frac{(1-p)\tilde{c}\mu_1^2}{3k_B T}H$, which can be identified with a magnetic susceptibility times $H$. The PDF $f(\mu) = (1-p)\delta(\mu - \mu_1) + pg(\mu)$ corresponds to a mixture of a single moment distribution and a continuous one. The meaning of the first distribution might be

associated with the sum of the magnetic moments of the randomly oriented ferrimagnetic particle cores [54].

In general, deviations from Langevin behavior can be caused by interparticle interactions, anisotropy and inhomogeneity such as volume and moment distribution. In fact, the distribution must in most cases lead to correct average parameters (even with narrow distributions). Furthermore, using a distributed function (e.g., a law with $T^{-1}$ or $T^{-1/2}$) in a distributed system with an inherent constant magnetization curve and an increase in the magnetic moment with temperature improves the consistency with phenomenological results. In fact, the choice of the distribution function is valuable in order to obtain a suitable parameter for the temperature variation, and separation of ferrimagnetic/paramagnetic and superparamagnetic components in a nanoparticles ensemble. However, it is not so critical initially considering any specific law or distribution function for the superparamagnetic part. In real systems, surface disorder, frustration, and rotational tilt can affect the moment distribution of the volume differently. The number of iron ions involved in superparamagnetism is clearly larger than in other magnetic orderings. The same is observed for the range between fully compensated and fully uncompensated configurations. In summary, using $T^{-\alpha}$ ($\alpha \neq 1$) implies that the uncompensated spins are not only on the surface, but also randomly distributed throughout the volume.

From now on, we will consider the particular model (see Appendix A)

$$M = c \int_0^\infty \mu L\left(\frac{\mu H}{k_B T}\right) f(\mu)\, d\mu \,. \tag{11}$$

This formula tells us that the magnetization is a function of $H/T$, regardless of the definition of $f(\mu)$. Hence, if we have experimental curves $M$ vs. $H$ for several temperatures, all curves will collapse into a single curve $M$ vs. $H/T$. Moreover, since the Langevin function is continuous and monotonic increasing, the magnetization of the system is also a continuous and monotonic increasing function $H/T$ and, consequently, for each value of $M$ there is a unique value of $H/T$. Therefore, if we have experimental curves of $M$ vs. $H$ for several temperatures, the curve $H$ vs. $T$ obtained by choosing the values of $H$ and $T$ for a fixed value of the magnetization will be a straight line.

Let us suppose that a particular magnetization curve follows exactly Equation (11). We now describe a method for obtaining an approximation of the magnetic moment distribution in this ideal case. First of all, we recall from probability theory that the cumulative distribution function (CDF) $F(\mu)$ associated with the PDF $f(\mu)$ that appears in Equation (10) given by $F(\mu) = \int_0^\mu f(x)dx$ [61]. The method consists in approximation of $F(\mu)$ by the step function

$$F_{A,n}(\mu) = \frac{1}{\sum\limits_{j=1}^{n} a_j} \sum_{k=1}^{n} a_k I\left(\mu - \frac{Ak}{n}\right), \tag{12}$$

where $A$ and $n$ are parameters, whose values must be chosen conveniently, $a_1, \ldots, a_n$ are unknowns to be determined, and $I(x) = 1$ if $x \geq 0$ and $I(x) = 0$ if $x < 0$. The values of $a_1, \ldots, a_n$ are obtained by solving (at least approximately) the following linear system of equations

$$M(H_i) = \sum_{k=1}^{n} a_k \frac{Ak}{n} L\left(\frac{AkH_i}{nk_B T}\right), \tag{13}$$

where $M(H_i)$ is the experimental value of the magnetization for a particular value of the magnetic field $H_i$ and $i$ runs over all the points of the magnetization curve. We notice that the linear system (Equation (13) ) can be inconsistent (indeterminate); for instance, when $n$ is less (greater) than the number of points of the magnetization curve. However, in any case, using the Moore–Penrose pseudoinverse [62,63] of the matrix whose $(i,k)$-term is $c_{ik} = \frac{Ak}{n} L\left(\frac{AkH_i}{nk_B T}\right)$, we can always obtain a unique vector $(a_1, \ldots, a_n)$ that minimize the sum

$$\sum_i \left[ M(H_i) - \sum_{k=1}^{n} a_k \frac{Ak}{n} L\left(\frac{AkH_i}{nk_BT}\right) \right]^2 \tag{14}$$

and has the minimum (Euclidean) norm. After obtaining good values for $a_1$, ..., $a_n$, in the sense that $F_{A,n}(\mu)$ approximates nicely to a monotonic nondecreasing function we take a numerical derivative of $F_{A,n}(\mu)$ to obtain an approximation of the probability distribution $f(\mu)$. For example, the difference quotient $\frac{n}{A}\left[F_{A,n}\left(\frac{Ak}{n}\right) - F_{A,n}\left(\frac{A(k-1)}{n}\right)\right] = \frac{na_k}{A\sum_{i=1}^{n} a_i}$ can be considered as an approximation to the derivative of $F_{A,n}(\mu)$ at the intermediate point $\mu = \frac{(2k-1)A}{2n}$ for $k = 1$, ..., $n$.

We apply the method described in the last paragraph in an example. Let us suppose we have a magnetization curve obtained by evaluating Equation (11) for

$$f(\mu) = \frac{3 \times 10^5 \sqrt{\mu}}{\pi\left(10^{10} + \mu^3\right)} , \tag{15}$$

at the magnetic field values $H_i = 400i$ with $i = 0, \pm 1, \pm 2, ..., \pm 50$. We chose the PDF given in Equation (15) because its shape is similar to the ones of the lognormal, gamma and Weibull PDFs, which were used in some of the models described in Section 2.2. Keeping $A = 10^5 \mu_B$ and considering several values of $n < 101$ (we have 101 points in the magnetization curve), we find that the graph of the step function $F_{A,n}(\mu)$, defined in Equation (12), shows significant oscillations. The reason for this is that $a_1$, ..., $a_n$ assume values that oscillate between positive and negative numbers. Fortunately, for values of $n \geq 400$, the graph of $F_{A,n}(\mu)$ stabilizes. For example, if $n = 500$, $F_{A,n}(\mu)$ approximates nicely to the CDF $F(\mu) = \int_0^{\mu} f(x) \, dx$ (see Figure 3). Taking a numerical derivative of $F_{A,n}(\mu)$, we obtain a good approximation to the PDF $f(\mu)$ (see Figure 4). If we consider other PDFs (for instance, the lognormal, gamma and Weibull ones) instead of the one given in Equation (15), we obtain analogous results after conveniently choosing the values of $A$ and $n$.

Unfortunately, in the case of a real magnetization curve, the procedure that has been shown to be useful in the ideal case does not work properly now. The main problem is that, for many values of $A$ and $n$ (even with n greater than the number of points in the magnetization curve), the values of $a_1$, ..., $a_n$ in the step function $F_{A,n}(\mu)$, defined in Equation (12), oscillate wildly between positive and negative values. Consequently, it is not possible to conclude whether $F_{A,n}(\mu)$ converges to a CDF, which is by definition monotonic nondecreasing.

A further study reveals that little coercivity gives rise to difficulties for the method presented in this section. However, if a non-uniform discretization is used in Equation (12), it is possible to elaborate another method following the same lines of the original one that resolves the difficulties. Unfortunately, the method and its variant cannot deal with the presence of noise in the magnetization curve, even in small quantities. For this reason, an improvement of this method is necessary in order to deal with the case of a real magnetization curve.

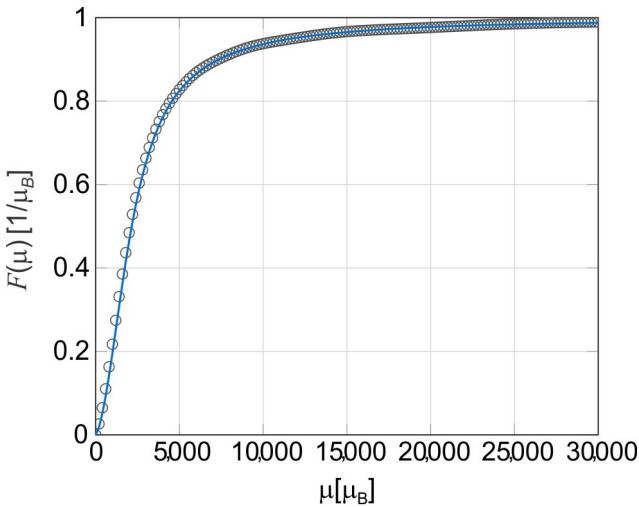

**Figure 3.** Representation of the cumulative distribution function $F(\mu)$ (continuous solid line) associated to the probability density function $f(\mu)$, defined in Equation (15), and the step function $F_{A,n}(\mu)$ with $A = 10^5\,\mu_B$ and $n = 500$, defined in Equation (11), which approximates to $F(\mu)$.

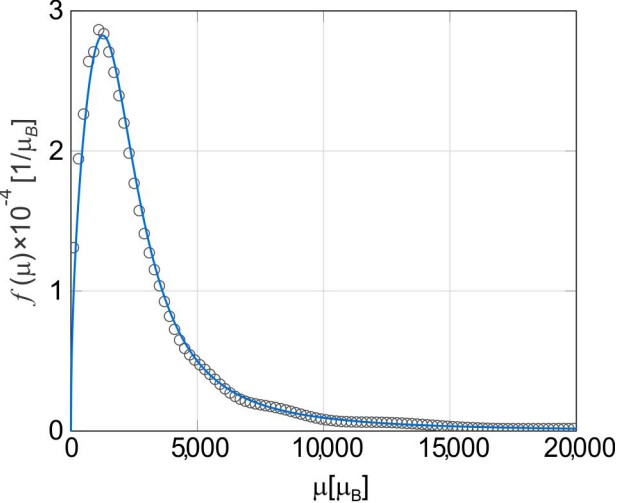

**Figure 4.** Representation of the probability density function $f(\mu)$ (continuous solid line), defined in Equation (15), and its approximation, obtained from taking a numerical derivative of $F_{A,n}(\mu)$ with $A = 10^5\mu_B$ and $n = 500$.

In summary, knowing the size distribution is very important to understanding the dynamic behavior of these systems. The examination of the magnetization curves is thus a suitable tool to get an idea about the particle size distribution and to resolve changes in the distribution. The extraction of the moment distribution function is performed by assuming some continuous distribution function such as, e.g., the gamma or lognormal distribution with adjustable parameters. The distribution function is then obtained by fitting the corresponding magnetization curve to the measured one. In this context, our results are similar to those presented by Rehberg et al. [64] where they used a method to reveal the characteristic magnetic moments of nanoparticles from their magnetization curves, using gamma and lognormal distributions. Notwithstanding, the Weibull distribution has been used in modeling lifetime data with monotonic failure rates. As far as we know, the use of this function to study the distribution of magnetic moments is an innovative contribution of our paper, expanding the possibilities of discussion about the effects of distributions on the magnetic properties of nanostructured systems.

## 4. Conclusions

We conclude that, in the investigation of the magnetic moment distribution in a system of magnetic nanoparticles (MNPs), the magnetization curve can be fitted almost equally well using the models (M), (L), (G) and (W) defined in Section 2.2. In particular, we have verified that the model (L), which involves a lognormal probability density function (lognormal PDF), is not the only one that can be used in the study of the magnetic moment distribution, i.e., other models can be considered; for instance, the models (G) and (W), which involve a gamma and Weibull PDFs. However, the shape of the obtained PDFs used in the models (L), (G) and (W) and their mean values are sometimes very different (for example, see the case of the Mag300-1h sample in Table 1 and Figure 2). This ambiguity raises the question of the actual magnetic moment distribution.

Fittings to the magnetization curves of all the samples considered in this article with the model given in Equation (10) are possible to be performed for PDFs of the form $f(\mu) = (1 - p)\delta(\mu - \mu_1) + pg(\mu)$, $0 \leq p \leq 1$. Replacing $g(\mu)$ with the lognormal, gamma, or Weibull PDFs, we obtain fits that are as good as the ones obtained with the models (L), (G) and (W). In some cases (for instance, for the Mag300-1h sample), the value of the parameter p differs very little from zero. Consequently, the continuous term $pg(\mu)$ in the expression of $f(\mu)$ has little relevance, compared to the discrete term $p\delta(\mu - \mu_1)$. This may justify the fact that the fitted curves obtained with the models (L), (G) and (W) are almost the same, although using PDFs with different shapes.

In Section 3 we proposed a method of obtaining the actual magnetic moment distribution directly from the magnetization curve when it follows exactly Equation (10). In this case, the distribution of points in the magnetization curve does not affect the effectiveness of the method. However, the presence of coercivity or noise in the magnetization curve gives rise to difficulties in the method. We believe that an improved version of this method would answer the open question on the magnetic moment distribution in a system of MNPs.

As a final remark, it can be pointed out that the models and procedures adopted in this work can be applied to study other systems that present nanomagnetism and, consequently, unconventional behaviors such as superparamagnetism. In this work, we considered an assembly of ferrimagnetic nanoparticles, but this study can be extended to nanosystems with other magnetic orderings as ferromagnetic or antiferromagnetic ones.

**Author Contributions:** Conceptualization, R.C.N., R.d.S.M. and L.F.C.; Formal analysis, M.J.J.R., D.S.V., R.C.N. and G.S.D.; Funding acquisition, L.F.C.; Investigation, M.J.J.R., D.S.V., G.S.D. and I.A.d.S.; Methodology, R.C.N. and L.F.C.; Project administration, L.F.C.; Supervision, R.d.S.M. and L.F.C.; Validation, I.A.d.S.; Writing—original draft, M.J.J.R. and D.S.V.; Writing—review & editing, I.A.d.S., R.d.S.M. and L.F.C. All authors have read and agreed to the published version of the manuscript.

**Funding:** This research was funded by Coordenação de Aperfeiçoamento de Pessoal de Nível Superior: Fellowship funding; Conselho Nacional de Desenvolvimento Científico e Tecnológico: Project Funding and fellowship funding; Fundação Araucária de Apoio ao Desenvolvimento Científico e Tecnológico do Estado do Paraná: Project Funding; Financiadora de Estudos e Projetos: Project Funding.

**Acknowledgments:** The authors would like to thank H. V. Ribeiro for fruitful discussions. Partial financial support from CAPES and CNPq (Brazilian agencies) is also acknowledged.

**Conflicts of Interest:** The authors declare no conflict of interest.

## Appendix A. Formula for the Magnetization

A simple model that takes into account the fact that the MNPs have different values of magnetic moment is based on the Hamiltonian

$$\mathcal{H} = -\sum_{i=1}^{N_1} \mu_1 H \cos\theta_i - \sum_{i=N_1+1}^{N_1+N_2} \mu_2 H \cos\theta_i - \cdots - \sum_{i=N_1+\cdots+N_{r-1}+1}^{N} \mu_n H \cos\theta_i , \qquad \text{(A1)}$$

where $\mu_1, \cdots, \mu_r$ are the possible values of the magnetic moment of each particle, $N_1, \cdots, N_r$ are the number of particles with magnetic moment $\mu_1, \cdots, \mu_r$, respectively, $(N_1 + \cdots + N_r = N)$ and $\theta_i$ is the angle between the $i$th magnetic moment and the external field **H**. The partition function associated to this Hamiltonian is

$$Z = \prod_{i=1}^{r} \left( \int_0^{2\pi} d\phi \int_0^{\pi} e^{\mu_i H \cos \frac{\theta}{k_B T}} \sin \theta \, d\theta \right)^{N_i} = \prod_{i=1}^{r} \left[ \frac{4\pi k_B T}{\mu_i H} \sin h \left( \frac{\mu_i H}{k_B T} \right) \right]^{N_i} \tag{A2}$$

and the total magnetization of the system is

$$\mathcal{M} = k_B T \frac{\partial \ln Z}{\partial H} = \sum_{i=1}^{r} N_i \left[ \mu_i coth \left( \frac{\mu_i H}{k_B T} \right) - \frac{k_B T}{H} \right] . \tag{A3}$$

Consequently, the mass magnetization is given by

$$M = c \sum_{i=1}^{r} p_i \mu_i L \left( \frac{\mu_i H}{k_B T} \right) , \tag{A4}$$

where $c = N = m$ ($m$ = total mass), $L(x) = coth x - 1/x$ is the Langevin function and $p_i = N_i = N$ is the probability of choosing a particle with a value of magnetic moment $\mu_i$. Thus, Equation (A4) says that the magnetization of the system is the average of the magnetizations of all MNPs. Therefore, if the number of the possible values for the magnetic moment of each particle, $r$, is very large, Equation (A4) may be approximated by

$$M = c \int_0^{\infty} \mu L \left( \frac{\mu H}{k_B T} \right) f(\mu) \, \mathrm{d}\mu , \tag{A5}$$

where $f(\mu)$ is a probability density and $f(\mu)d\mu$ is the probability of choosing a particle with a value of magnetic moment in the interval $[\mu, \mu + d\mu]$. It should be also noticed that Equation (A5) leads to Equation (A4) if we admit

$$f(\mu) = \sum_{i=1}^{r} p_i \delta(\mu - \mu_i), \quad \sum_{i=1}^{r} p_i = 1 , \tag{A6}$$

as a probability density, where $\delta(x)$ is the Dirac delta distribution.

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
