# Peer review of "On the Distribution of Magnetic Moments in a System of Magnetic Nanoparticles"

_magnetochemistry, doi:10.3390/magnetochemistry8100129_

Round 1

Reviewer 1 Report

The manuscript deals with a critical analysis of several currently used models for the description of probability density function (PDF) of magnetic moments in magnetic nanoparticles. The authors find that several known PDF models describe the distribution of magnetic moments equally well and cannot be discriminated by a mere fitting of experimental magnetization curves. Concomitantly they propose a new model and a method for obtaining the actual distribution of magnetic moments. The obtained results are important for a broad readership of Magnetochemistry. The article is well written, so I recommend its publication as is.

Author Response

The manuscript deals with a critical analysis of several currently used models for the description of probability density function (PDF) of magnetic moments in magnetic nanoparticles. The authors find that several known PDF models describe the distribution of magnetic moments equally well and cannot be discriminated by a mere fitting of experimental magnetization curves. Concomitantly they propose a new model and a method for obtaining the actual distribution of magnetic moments. The obtained results are important for a broad readership of Magnetochemistry. The article is well written, so I recommend its publication as is.

Comments: 

Dear Reviewer,

We would like to thank you for the revision on our submitted manuscript ‘‘On the distribution of magnetic moments in a system of magnetic nanoparticles’’ (magnetochemistry-1866595). The manuscript was improved following the questions from the other reviewers.

Reviewer 2 Report

Many researchers use a lognormal probability density function to describe the particle size and magnetic moment distributions in magnetic nanoparticles assemblies as magnetite and maghemite. Jauregui et al. showed that the lognormal probability density function is NOT The ONLY ONE that can be used in the study of the magnetic moment distribution. And the author et al. investigated a method for identifying the actual magnetic moment distribution in a system of magnetic nanoparticles. However, the presence of coercivity or noise in the magnetization curve gave rise to difficulties. The results are well organized and are worth publishing.

Author Response

Dear Reviewer,

We would like to thank you for the revision on our submitted manuscript ‘‘On the distribution of magnetic moments in a system of magnetic nanoparticles’’ (magnetochemistry-1866595). The manuscript was improved following the questions from the other reviewers.

Reviewer 3 Report

1. The title is not very suggestive.

2. Abstract should clearly inform the important findings in the present study

3. The lengthy sentences may be split in to smaller sentence without change of its meaning.

4. Also, suggested to include the recent references in the introduction part.

5. Magnetic nanoparticles (MNPs) have been used in numerous technological applications. A variety of these particles, composed of distinct atoms or ions with different magnetic moments, have been synthesized to be applied in many research fields.Compare the properties of MNPs obtained from the literature. You can use the references: Journal of thermal analysis and calorimetry 97 (1), 2009, 245-250; Acta Chim. Slov. 2009, 56, 379–385 and  Journal of thermal analysis and calorimetry 94 (2), 2008, 389-393

6. The author have detailed the experimental part, in its current form it is not very clear. Insert a table with the number of moles of all reagents used in the synthesis.

7. The authors should correlate their performance results with the already published studies of different researchers to show the priority of their research study.

8 The results and discussions part should be compared with the literature data. To redo the part of results and discussions by a systematic presentation of the results by which the readers of the articles manage to follow the article more easily.

9. Go through all the hysteresis curves for all the compounds studied in figure 1. They could be superimposed to better see the differences.

10. In discussions about structural analysis, XRD and XPS measurements are missing. Let these measurements be added to have a substrate for the correlation of the discussions.

11. Conclusions should be short with important observations.

12. At the conclusion, do not pass the graphs, move the graphs and the discussion to the results and discussions. Let only the concluded aspects remain in the conclusions.

13. References are not written in unison. Some journals are abbreviated and some are not.

14. The authors must revise language of the manuscript before publication and the whole article should be adjusted based on journal style.

Author Response

Dear Reviewer,

We would like to thank you for the revision on our submitted manuscript ‘‘On the distribution of magnetic moments in a system of magnetic nanoparticles’’ (magnetochemistry-1866595). The questions were carefully analyzed, and the manuscript was modified (improved). Please find below our specific answers.

1. The title is not very suggestive.

We thank the reviewer for the comment. We tried our best to get a more suggestive title, but we still think this is the best title for this manuscript.

2. Abstract should clearly inform the important findings in the present study

We thank the reviewer for the comment. We improved the abstract.

3. The lengthy sentences may be split in to smaller sentence without change of its meaning.

We thank the reviewer for the comment. We improved the text in the whole manuscript.

4. Also, suggested to include the recent references in the introduction part.

We thank the reviewer for the comment. New recent references were added to the manuscript.

5. Magnetic nanoparticles (MNPs) have been used in numerous technological applications. A variety of these particles, composed of distinct atoms or ions with different magnetic moments, have been synthesized to be applied in many research fields.Compare the properties of MNPs obtained from the literature. You can use the references: Journal of thermal analysis and calorimetry 97 (1), 2009, 245-250; Acta Chim. Slov. 2009, 56, 379–385 and Journal of thermal analysis and calor…

New references were added to the manuscript.

Stefanescu, M.; Stoia, M.; Caizer, C.; Dippong, T.; Barvinschi, P. Preparation of CoxFe3−xO4 nanoparticles by thermal decomposition of some organo-metallic precursors. Journal of Thermal Analysis and Calorimetry 2009, 97, 245-250.

Stefanescu, M.; Stoia, M.; Dippong, T.; Stefanescu, O.; Barvinschi, P. Preparation of CoXFe3–XO4 Oxydic System Starting from Metal Nitrates and Propanediol. Acta Chimica Slovenica 2009, 56, 379-385.

6. The author have detailed the experimental part, in its current form it is not very clear. Insert a table with the number of moles of all reagents used in the synthesis.

We thank the reviewer for the comment. But, in our opinion, this informations do not make sense in this manuscript.

7. The authors should correlate their performance results with the already published studies of different researchers to show the priority of their research study.

Know the size distribution is very important to understand the dynamic behavior of this systems. The examination of the magnetization curves is thus a suitable tool to get an idea about the particle size distribution and to resolve changes of the distribution.

The extraction of the moment distribution function is done by assuming some continuous distribution function like, e.g., the gamma or log-normal distribution with adjustable parameters. The distribution function is then obtained by fitting the corresponding magnetization curve to the measured one. In this context, our results are similar those presented by Ingo Rehberg et al. where they used a method to reveal the characteristic magnetic moments of nanoparticles from their magnetization curves, using gamma and log-normal distributions. Notwithstanding, the Weibull distribution has been used in modeling lifetime data with monotonic failure rates. As far as we know, the use of this function to study the distribution of magnetic moments is an innovative contribution of our paper, expanding the possibilities of discussion about the effects of distributions on the magnetic properties of nanostructured systems.

Rehberg, I., Richter, R., Hartung, S., Lucht, N., Hankiewicz, B., & Friedrich, T. (2019). Measuring magnetic moments of polydisperse ferrofluids utilizing the inverse Langevin function. Physical Review. B, 100(13). https://doi.org/10.1103/physrevb.100.134425

8. The results and discussions part should be compared with the literature data. To redo the part of results and discussions by a systematic presentation of the results by which the readers of the articles manage to follow the article more easily.

We thank the reviewer for the comment. But, in our opinion, the format presented in the manuscript can lead the reader in a better way to understand the findings.

9. Go through all the hysteresis curves for all the compounds studied in figure 1. They could be superimposed to better see the differences.

We thank the reviewer for the comment. The figure 1 has the goal to show/highlight the best fit for each hysteresis curve, not to do a comparison between them.

10. In discussions about structural analysis, XRD and XPS measurements are missing. Let these measurements be added to have a substrate for the correlation of the discussions.

We thank the reviewer for the comment. However, there is no such discussion of structural analysis in this manuscript. In fact, this is not the focus of the work and the lack of this discussion does not harm it.

11. Conclusions should be short with important observations.

We thank the reviewer for the comment. But, in our opinion, the format presented in the Conclusions can lead the reader in a better way to understand our findings.

12. At the conclusion, do not pass the graphs, move the graphs and the discussion to the results and discussions. Let only the concluded aspects remain in the conclusions.

We thank the reviewer for the comment. We moved the figure to the right place.

13. References are not written in unison. Some journals are abbreviated and some are not.

We thank the reviewer for the comment. We improved the references.

14. The authors must revise language of the manuscript before publication and the whole article should be adjusted based on journal style.

We thank the reviewer for the comment. We improved the text in the whole manuscript.

Reviewer 4 Report

The authors present a study on using different magnetic moment distributions of magnetic nanoparticles (MNPs). They have synthesized magnetite MNPs that they test with these distributions and models on (M/H-analysis) and discuss the result. The manuscript needs to be revised so it will be easier to read. Suggestions and comments to the authors are given below.

Abstract: “Then, there is no …”. Strange sentence, what is the message from the authors. Rephrase.

Introduction: First sentences. add also the paper by Q. Pankhurst et al Journal of physics D: Applied physics 36 (13), R167 (2003) where more biomedical applications are presented and discussed.

Introduction, 2nd page: add also the papers by Bender et al New Journal of Physics 19 (7), 073012 (2017) and Berkov D V, Görnert P, Buske N, Gansau C, Mueller J, Giersig M, Neumann W and Su D, J. Phys. D: Appl. Phys. 33 331, 2000, where general distribution functions are used in different analysis methods for magnetic nanoparticles.

Section 2.1: Have the authors measured particle sizes of the used samples? For instance, TEM for the magnetic core in the particle. Add magnetic core size data of the synthesized particles or refer to paper(s) where this information is included. Also, other analysis methods XRD (or Mossbauer) for analysing the material properties. All of this this is very important when comparing the result from the M/H fitting.

Section 2.2. Several comments. a) add in the text that the Langevin function assumes no magnetic interactions and no magnetic anisotropy energy contribution, b) the linear field part in M is used for the high field part of the M/H curve (gives the high field susceptibility). Add some reference of this. This high field susceptibility is used for (for instance) the spin canting effect or some paramagnetic contribution to M. Ref 45 is for antiferromagnetic nanoparticles that is not the case for these magnetite nanoparticles (ferrimagnetic). Add more explanation here. c) In all of the presented models, add that lambda is the median core size and beta gives the width of the distribution.

Section 2.3. Three comments. a) what is causing the large relative high field susceptibility for sample MAG300-1h (figure 1 top left). This M/H loop is not usual for magnetite MNPs., b) Figure 1; Mass magnetization. What is the mass the measured sample magnetic moment is divided with, sample mass or total mass of the particles? c) have the authors subtracted the sample holder signal (usually a diamagnetic contribution).

Section 3. row 156: why this T^(-1/2) behaviour? is it due to interactions? sometimes there could also be a Curie-Weiss law (low interaction regime). Add more explanation here.

Section 3. row 159: Two comments. a) why using this distribution function? and a general comment; why including this discussion and analysis that ends with eq. 9? More explanations here. b) What is the physics behind this function? It looks like a superposition of distributions with only one single moment and a distribution of moments.

Section 3. row 167: “… collapse on a single curve …”. This is true if the nanoparticles are non-interacting and no magnetic anisotropy contributions.

Section 3. row 173: ok, so now the authors use a model without the linear field term (the high field susceptibility) as in eq. 8. Is this the case?

Section 3. Figure 2. why does not the W and G models give a reasonable distribution for sample MAG300-1h as seen in figure 2 top left? Is it due to the large relative high field susceptibility?

Section 3. Equation 14. From where comes this function? Add more explanation here and change the text so it is easier for the reader to follow.

Section 3. Row 217. Instead write “low coercivity”. This is expected since the Langevin function does not show coercivity (or remanence).

Conclusions. Update this section so it agrees with the rest of the manuscript after the revision.

Author Response

It looks like these are same comments as for Reviewer 3. As follows, we are sending the same responses here.

Dear Reviewer,

We would like to thank you for the revision on our submitted manuscript ‘‘On the distribution of magnetic moments in a system of magnetic nanoparticles’’ (magnetochemistry-1866595). The questions were carefully analyzed, and the manuscript was modified (improved). Please find below our specific answers.

- The authors present a study on using different magnetic moment distributions of magnetic nanoparticles (MNPs). They have synthesized magnetite MNPs that they test with these distributions and models on (M/H-analysis) and discuss the result. The manuscript needs to be revised so it will be easier to read. Suggestions and comments to the authors are given below.

- Abstract: “Then, there is no …”. Strange sentence, what is the message from the authors. Rephrase.

We thank the reviewer for the comment. We improved the text.

- Introduction: First sentences. add also the paper by Q. Pankhurst et al Journal of physics D: Applied physics 36 (13), R167 (2003) where more biomedical applications are presented and discussed.

We thank the reviewer for the comment. We added the reference.

- Introduction, 2nd page: add also the papers by Bender et al New Journal of Physics 19 (7), 073012 (2017) and Berkov D V, Görnert P, Buske N, Gansau C, Mueller J, Giersig M, Neumann W and Su D, , where general distribution functions are used in different analysis methods for magnetic nanoparticles.

We thank the reviewer for the comment. We added the references.

- Section 2.1: Have the authors measured particle sizes of the used samples? For instance, TEM for the magnetic core in the particle. Add magnetic core size data of the synthesized particles or refer to paper(s) where this information is included. Also, other analysis methods XRD (or Mossbauer) for analysing the material properties. All of this this is very important when comparing the result from the M/H fitting.

TEM and XRD studies using the same nanoparticles as in this manuscript are well describe in the already cited Ref. [36]. 

- Section 2.2. Several comments. 

  1. a) add in the text that the Langevin function assumes no magnetic interactions and no magnetic anisotropy energy contribution, 

We thank the reviewer for the comment. We improved the text.

  1. b) the linear field part in M is used for the high field part of the M/H curve (gives the high field susceptibility). Add some reference of this. This high field susceptibility is used for (for instance) the spin canting effect or some paramagnetic contribution to M. Ref 45 is for antiferromagnetic nanoparticles that is not the case for these magnetite nanoparticles (ferrimagnetic). Add more explanation here. 

The ferrimagnetic state has the same spin orientation as the antiferromagnetic state. That is the reason we choose to follow ref. 45. 

But, of course, the effect can come from some paramagnetic contribution to M., i. e., the mass magnetic susceptibility χ can take on the role of different magnetic configurations/states, including some exotic states. As an example, we can cite the work of Li et al.  We'll add this comment to the manuscript.

Li, C.-Y., Karna, S. K., Wang, C.-W., & Li, W.-H. (2013). Spin polarization and quantum spins in Au nanoparticles. International Journal of Molecular Sciences, 14(9), 17618–17642. https://doi.org/10.3390/ijms140917618

  1. c) In all of the presented models, add that lambda is the median core size and beta gives the width of the distribution.

We thank the reviewer for the comment. We improved the text.

- Section 2.3. Three comments. a) what is causing the large relative high field susceptibility for sample MAG300-1h (figure 1 top left). This M/H loop is not usual for magnetite MNPs., 

As can be seen in the manuscript, this is the sample with the smallest particle size. Consequently, it has a lower blocking temperature. In this case below room temperature. The shape of the curve is typical of ferrimagnetic nanoparticles in the blocked state.

  1. b) Figure 1; Mass magnetization. What is the mass the measured sample magnetic moment is divided with, sample mass or total mass of the particles? 

Typically, we used about 50 mg of sample in our experiments.

  1. c) have the authors subtracted the sample holder signal (usually a diamagnetic contribution).

The sample holder's response is negligible.

- Section 3. row 156: why this T^(-1/2) behaviour? is it due to interactions? sometimes there could also be a Curie-Weiss law (low interaction regime). Add more explanation here.

Deviations from Langevin behavior can be caused by interparticle interactions, anisotropy and

inhomogeneity such as volume and moment distribution. In fact, the distribution must be

in most cases leading to correct average parameters (even with narrow distributions).

Furthermore, using a distributed function (e.g. a law with T^-1 or T^-1/2) in a distributed system with an inherent constant magnetization curve and an increase in the magnetic moment with temperature. In addition, the choice of the distribution function is so critical in order to obtain a suitable parameter for the temperature variation, separation of ferrimagnetic/paramagnetic and superparamagnetic components in nanoparticles ensemble without initially considering any specific law or distribution function for the superparamagnetic part can be considered.

In real systems, surface disorder, frustration, and rotational tilt can affect different moment distributions of volume. The number of iron ions involved in superparamagnetism is clearly larger and the range between fully compensated and fully uncompensated configurations. In summary, using T^-1/2 implies that the uncompensated spins are not only on the surface, but also randomly distributed throughout the volume. [according to Ref. 43]

- Section 3. row 159: Two comments. a) why using this distribution function? and a general comment; why including this discussion and analysis that ends with eq. 9? More explanations here. 

As already mentioned, magnetic nanoparticle systems are remarkably complex. In some specific cases, scaling laws are sufficient to adequately explain the behavior of the system. In other cases, distribution functions are undoubtedly required. A third reasonable possibility requires a proper mixture between scaling laws and distribution functions. Our goal in this work is to shed light on this matter and contributes to the discussion and perhaps offers an alternative.

  1. b) What is the physics behind this function? It looks like a superposition of distributions with only one single moment and a distribution of moments.

The explanation is in the previous answers. We are undoubtedly working with very complex systems and therefore, sometimes, the satisfactory answers are still not simple.

General comments to Section 3 Row 159:  a-b) The discussion that ends in Eq. (10) (old Eq. (9)) serves to show that, when the mass magnetic susceptibility follows a T^{-1} law, Eq. (9) (old Eq. (10)) contains Eq. (8) as a particular case. This is explicitly shown considering a magnetic moment distribution which is a mixture of a single moment distribution and a continuous one.

- Section 3. row 167: “… collapse on a single curve …”. This is true if the nanoparticles are non-interacting and no magnetic anisotropy contributions.

We thank the reviewer for the comment.

- Section 3. row 173: ok, so now the authors use a model without the linear field term (the high field susceptibility) as in eq. 8. Is this the case?

The aim of our manuscript is to present new possibilities for the study of nanostructured magnetic systems. This is a possibility studied and discussed. In other words, we intend to provide tools for the most different points of view, since the study of these nanosystems is very complex and can generate different interpretations.

- Section 3. Figure 2. why does not the W and G models give a reasonable distribution for sample MAG300-1h as seen in figure 2 top left? Is it due to the large relative high field susceptibility?

As addressed in the previous response, the aim of our manuscript is to present new possibilities for the study of nanostructured magnetic systems. For the case of sample MAG300-1h it is clear that G and W models are not able to describe the system very well. That is, other alternatives are necessary for a better understanding of the system.

- Section 3. Equation 14. From where comes this function? Add more explanation here and change the text so it is easier for the reader to follow.

The explanation about this function is in between lines 173 and 194. The numeric values are explained in the text.

- Section 3. Row 217. Instead write “low coercivity”. This is expected since the Langevin function does not show coercivity (or remanence).

We thank the reviewer for the comment. We improved the text.

- Conclusions. Update this section so it agrees with the rest of the manuscript after the revision.

We thank the reviewer for the comment. But, in our opinion, the format presented in the Conclusions can lead the reader in a better way to understand our findings.

Round 2

Reviewer 4 Report

The authors have revised their manuscript according to some of the previous comments. Still there are some comments and questions given below.

1. None of the suggested additional references are added to the reference list and the text, even if the authors state that in their response letter. It is good to add these references where a general distribution has been used (in these cases no assumption of a specific function is used in the fitting process). There are new references added but this is probably from the other reviewer comments?

2. The authors shall add the particle/core sizes in the manuscript from the TEM result in ref. 42 (the authors claims that the TEM data in in ref. 36 but that is a reference from 1980, Chantrell et al). These particle/core sizes shall be compared with the determined particle magnetic moment distribution functions using the particle magnetic moment and saturation magnetization to obtain the particle/core size. This is a very important way of assessing if the fitting result is good or not. The authors shall include this comparison in the manuscript and discuss the result.

3. The authors claim in the response letter that they have added that “ lambda is the median core size and beta gives the width of the distribution”, but this cannot be seen in the revised version of the manuscript. Add this to the manuscript. It will be easier for the reader.

4.  Regarding sample MAG300-1h and the comment/question on the large high field susceptibility in the M/Loop (figure 1). The authors claim that this is the sample with the smallest particle size (but this is nothing we can see from this manuscript) and also that the M/H curve for this sample is typical for magnetic nanoparticles in a blocked regime. This is not true. Thermally blocked magnetic nanoparticles will show hysteresis (remanence and coercivity) but this cannot be seen in the M/H-loop for MAG300-1h. The high field susceptibility cannot be due to thermal blocked properties. Instead, the high field susceptibility can be due to for instance a large spin canting effect (that may be due to that the particles has a small core size)? Add some discussions about this in the text.

5. Add the temperature in figure 1 (the sample temperature in the M/H-loop measurements).

6. For sample MAG300-1h the fitting result (i.e. the determined distribution function, figure 2 top left) is not good using the G and W model, even if these models contain a linear high field susceptibility. Please, give an explanation of this in the manuscript.

7. It can be good to also add in the figure 2 text that the particle magnetic moment on the x-axis is represented as Bohr magnetons (even if this stated in the table 1 text).

Author Response

Dear Reviewer,

We would like to thank you for revising our manuscript. The questions were carefully analyzed, and the manuscript was modified (improved). Please find below our specific answers.

  1. None of the suggested additional references are added to the reference list and the text, even if the authors state that in their response letter. It is good to add these references where a general distribution has been used (in these cases no assumption of a specific function is used in the fitting process). There are new references added but this is probably from the other reviewer comments?

We sincerely apologize for the possible mistake. Something got wrong in the final file. We again added the requests.

  1. The authors shall add the particle/core sizes in the manuscript from the TEM result in ref. 42 (the authors claims that the TEM data in in ref. 36 but that is a reference from 1980, Chantrell et al). These particle/core sizes shall be compared with the determined particle magnetic moment distribution functions using the particle magnetic moment and saturation magnetization to obtain the particle/core size. This is a very important way of assessing if the fitting result is good or not. The authors shall include this comparison in the manuscript and discuss the result.

We sincerely apologize for the mistake. Something got wrong in the final file. We added the right reference number.

The particle diameter sizes are between 15 and 20 nm (similar sizes were calculated via X-ray diffraction measurements). We included this information to the manuscript.

However, it is not the specific intention to show the size calculation (we have done this in other works). Here we want to draw attention to different ways of obtaining distributions and how complex the work to achieve the “most correct” distribution can be.

  1. The authors claim in the response letter that they have added that “ lambda is the median core size and beta gives the width of the distribution”, but this cannot be seen in the revised version of the manuscript. Add this to the manuscript. It will be easier for the reader.

We sincerely apologize for the mistake. Something got wrong in the final file. We added the sentence to the manuscript.

  1. Regarding sample MAG300-1h    he authors claim that this is the sample with the smallest particle size (but this is nothing we can see from this manuscript) and also that the M/H curve for this sample is typical for magnetic nanoparticles in a blocked regime. This is not true. Thermally blocked magnetic nanoparticles will show hysteresis (remanence and coercivity) but this cannot be seen in the M/H-loop for MAG300-1h. The high field susceptibility cannot be due to thermal blocked properties. Instead, the high field susceptibility can be due to for instance a large spin canting effect (that may be due to that the particles has a small core size)? Add some discussions about this in the text.

We thank the reviewer for the comment and discussion. We added to the manuscript this valuable contribution.

  1. Add the temperature in figure 1 (the sample temperature in the M/H-loop measurements).

We thank the reviewer for the comment. We added the temperature to the figure caption.